# Scope of health worker migration governance and its impact on emigration intentions among skilled health workers in Nigeria

**Kenneth Yakubu**[1]*, **Janani Shanthosh**[1,2], **Kudus Oluwatoyin Adebayo**[3,4], **David Peiris**[1], **Rohina Joshi**[5,6]

**1** The George Institute for Global Health, Faculty of Medicine, University of New South Wales, Sydney, Australia, **2** Australian Human Rights Institute, Faculty of Law and Justice, University of New South Wales, Sydney, Australia, **3** African Centre for Migration and Society, University of The Witwatersrand, Johannesburg, South Africa, **4** Diaspora and Transnational Studies Unit, Institute of African Studies, University of Ibadan, Ibadan, Nigeria, **5** School of Population Health, Faculty of Medicine, University of New South Wales, Sydney, Australia, **6** The George Institute for Global Health India, New Delhi, India

* kyakubu@georgeinstitute.org.au

**Data Availability Statement:** We have provided de-identified quantitative data within the supplementary files of this submission. Further requests from any qualified investigator can be

## Abstract

The growing trends for skilled health worker (SHW) migration in Nigeria has led to increased concerns about achieving universal health coverage in the country. While a lot is known about drivers of SHW migration, including national/sub-national government's inability to address them, not enough is known about its governance. Underpinning good governance systems is a commitment to human rights norms, that is, principles that enshrine non-discrimination, participation, accountability, and transparency. Hence, this study was aimed at deriving a conceptual framework that captures the scope of SHW migration governance in Nigeria and the extent to which it is human rights based. To describe the scope of SHW migration governance, we conducted an exploratory factor analysis and mapped our findings to themes derived from a qualitative analysis. We also did a multivariate analysis, examining how governance items are related to migration intentions of SHWs. The scope of SHW migration governance in Nigeria can be described across three levels: Constitutional —where policies about the economy and the health workforce are made and often poorly implemented; Collective—which responds to the governance vacuum at the constitutional level by promoting SHW migration or trying to mitigate its impact; Operational—individual SHWs who navigate the tension between the right to health, their right to fair remuneration, living/working conditions, and free movement. Examining these levels revealed opportunities for collaboration through stronger commitment to human right norms. In recognising their role as rights holders and duty bearers at various levels, citizens, health advocates, health workers, community groups and policy makers can work collaboratively towards addressing factors related to SHW migration. Further evidence is needed on how human rights norms can play a visible role in Nigeria's governance system for SHW migration.

addressed to the Head, Data Management; email: datamanagement@georgeinstitute.org.au. Excerpts from the transcribed qualitative data relevant to the study have been provided within the paper.

**Funding:** The authors received no specific funding for this work.

**Competing interests:** The authors have declared that no competing interests exist.

## Introduction

To achieve the health-related Sustainable Development Goals (SDGs), the World Health Organisation has recommended a skilled health worker (SHW) density of 4.45 per 1000 [1]. As at 2018, Nigeria had a SHW density of 1.83 per 1000 [2]. The factors contributing to the low density of SHWs include crisis in the educational sector leading to low production of an adequate health workforce, poor management/leadership within the health system, political, and economic crises leading to an increasing trend of migration of SHWs from Nigeria [3]. Between 2008 and 2021, a total of 36,467 Nigerian doctors migrated to the United Kingdom. There was a steady increase from 1,798 that migrated in 2008 to 4,880 in 2021. A larger trend was observed for nurses. Between 2002 and 2021, a total of 60,729 Nigerian nurses had migrated to the United Kingdom. There was a steady increase from 1,393 nurses that migrated in 2002, to 5,543 in 2021 [4].

There are concerns about the impact of SHW migration on Nigeria's health system. These impacts include an increase in the clinical workload, reduction in the quality of care given by SHWs who remain, and an increase in the mortality and morbidity patterns arising from a lack of access to essential health care services [5]. Increasing the number of SHWs through improved training and recruitment programs are important steps towards addressing the situation. However, increasing the number of SHWs in the health system will have negligible impact without addressing factors related to their retention and migration, hence the need to improve SHW migration governance [6].

The governance of migration has been defined as "the combined frameworks of legal norms, laws and regulations, policies, and traditions as well as organisational structures (subnational, national, regional, and international), and the relevant processes that shape and regulate States' approaches with regard to migration in all its forms, addressing rights and responsibilities and promoting international cooperation" [7]. Simply put, it refers to rules (formal or informal) and processes that shape how state and non-state actors respond to migration, foster cooperation between states, and uphold human rights [8]. These definitions apply to migration governance from a broad societal perspective, but its link to the health sector is important because of the unique role health systems play in protecting population health and wellbeing [9–11]. For this study, we relied on an institutional framing of governance (that focuses on the types of rules, informal or formal, that influence responses to demand and supply relations [12]) and defined SHW migration governance as a system of rules which exist to provide oversight of the SHW migration process, mitigate the impact of SHW migration on health service delivery, and that directly or indirectly influence SHW's intention to migrate. Inherent to this definition is our recognition of human rights norms as essential for promoting migration and health system governance systems that are people-centred [13, 14]. These norms include the right to migrate, fair wages, and proper working conditions for SHWs, while protecting access to essential health services and the preconditions for health.

In 2015, the Federal Government of Nigeria, with the support of international donors, developed a National Migration Policy [15]. This policy has recommendations for managing migration in Nigeria but does not yet address challenges associated with SHW migration. While previous studies in the Nigerian context have focused on the drivers of health worker migration [16–19] and the role of the government in addressing them [15, 20–22], there is a need to also understand the role of other stakeholders, exploring how collaboration might occur between them. Since SHWs are the primary endpoints of actions/interventions from both state and non-state actors, they are a rich source of information, describing formal and informal rules that influence their migration intentions. The description so gained will offer a close-to-the ground conceptualisation of how SHW migration governance occurs in Nigeria,

offering an opportunity to identify factors that can be leveraged to maintain an adequate number of SHWs. Hence, the aim of this study was to develop a relevant framework that captures the scope of SHW migration governance and explore its utility for influencing migration intentions of health professionals in Nigeria. The first objective was to develop a relevant framework that captures the scope of SHW migration governance in Nigeria and determine the extent to which health professional migration governance in Nigeria is human rights based. The second was to determine the relationship between the derived governance items and migration intentions of health professionals in Nigeria.

## Materials and methods

### Study design

We used a mixed-methods (concurrent) approach for this study–a cross-sectional survey and semi-structured interviews were collected and analysed during the same period [23]. We have provided a summary of the methods in Fig 1.

### Participants

We included nurses, dentists, pharmacists, and medical doctors since they tend to migrate more and are often the focus of many skilled health migration studies and policies [24–27]. We also included SHWs who resided and/or practiced in Nigeria, irrespective of how long they had been in the country, or their history of previous migration. Skilled health workers such as physiotherapists, podiatrists were excluded from this study as they are not often mentioned in the literature for SHW migration.

### Sample size and sampling technique

For the survey, our assumption for the sample size was informed by the requirements for an conducting an exploratory factor analysis (EFA). In Costello & Osborne's analysis of surveys

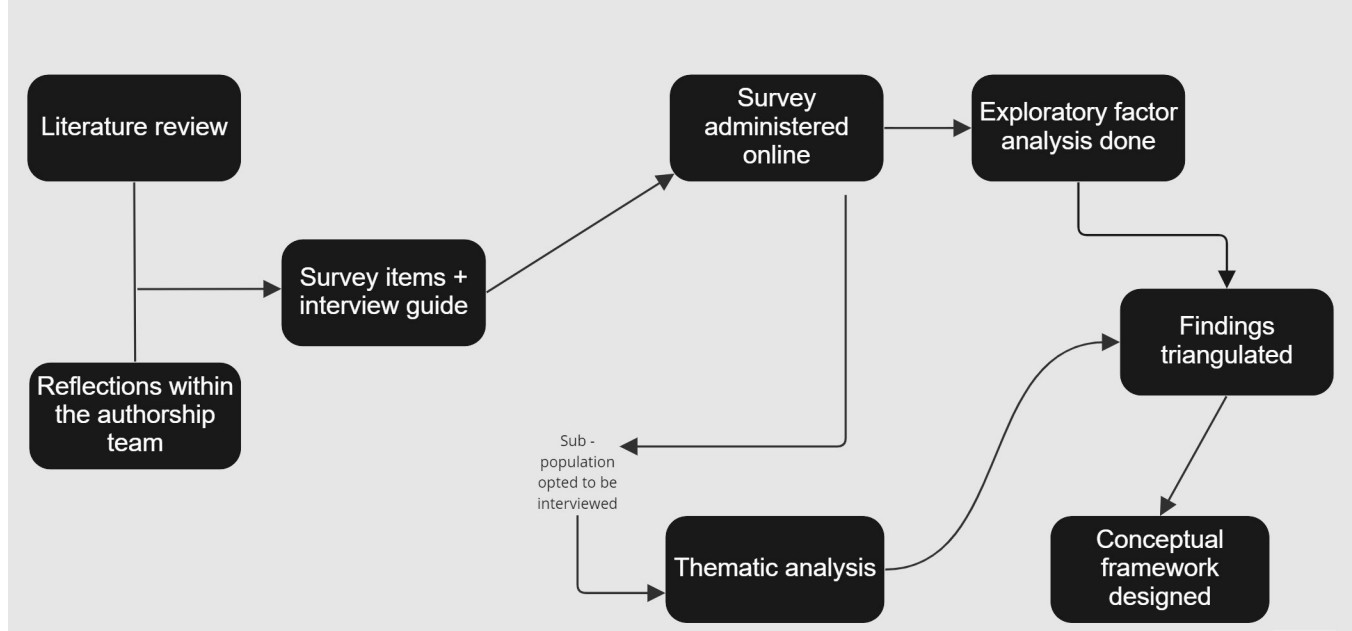

**Fig 1. A summary of the methods.**

that involved an EFA, at least 40% of the samples that used a 5:1 participant to item (P/I) ratio were able to achieve a correct factor structure. This increased to 70% of samples when the P/I ratio was increased to 20:1 [28]. Considering what was practical in our setting, we chose a 5:1 P/I ratio. With 38 items, this yielded a minimum sample size of 190. Allowing for a 32% response rate (based on a previous e-survey among health professionals [29, 30]), we arrived at a minimum sample size of 250 and estimated that we will be able to recruit 63 each of nurses, dentists, pharmacists, and doctors for the survey. We eventually recruited 271 participants using a convenience sampling technique.

For the qualitative interviews, we chose a purposive sampling approach and invited five participants from each professional group of nurses/midwives, dentists, pharmacists, and doctors. Even though no dentist opted in for the interviews, we achieved a total of 22 interview participants (including six nurses, six pharmacists, and ten medical doctors) and the interviews concluded when saturation was achieved [31–33].

## Study tool

KY conducted a non-systematic literature search in Google Scholar using the key words: "migration governance"; "rights-based" AND "health systems governance"; "migration of health professionals" AND "Nigeria". From the first three pages, thirteen articles that offered insights on drivers of migration in Nigeria, health system and migration governance, and rights-based approaches to governance were identified [12, 34–45]. Drawing on key thematic areas from across these sources and reflections on his lived experience, KY generated an initial list of 30 survey items relevant to SHW migration governance in Nigeria.

Following discussions with the co-authors, these were refined for clarity and ease of comprehension, and expanded to include other relevant items. The final survey tool had a list of 46 items, one described the outcome variable (i.e., migration intentions) and seven elicited demographic characteristics of the participants. Thirty-eight (38) items captured participants' agreement to statements that describe SHW migration governance in Nigeria. For example, some statements suggested how rules governing health professional migration were made, whether rights-based norms inform the handling of SHW migration, and the contributions of relevant stakeholders to SHW migration governance. The scoring of participants' responses to each item ranged from 1–5. Scores of 1 and 2 represented a poor perception of SHW migration governance, 3 represented a neutral view, and 4–5 represented a good perception. Since questions 1, 2, 7, and 32 suggested a negative governance performance, their values were reversed before the analysis. We identified migration intention as the dependent variable which comprised of two levels–staying and practicing versus migrating out of the country.

The interview guide for the semi-structured interviews included open-ended questions on the participant's experience working in Nigeria, their migration intentions, stakeholders in the governance of SHW migration, patterns of behaviour and interaction among stakeholders, and their recommendations for change. The survey tool and interview guide were pretested for clarity and ease of administration among three volunteer health professionals in Nigeria. Their feedback was useful for further refining them and did not require any item to be removed.

## Study procedure

We administered the survey by sending a web link to the social media platforms of four professional associations (nurses/midwives, pharmacists, dentists, and medical doctors) in Nigeria. We also distributed the survey link through the personal networks of individual health professionals, and it remained open from 10th August 2020 to 26th April 2021. After completing the

survey, participants were invited to be part of the semi-structured interviews. These interviews were conducted by KY using online video calls.

## Data analysis

For the first objective (which is to develop a relevant framework that captures the scope of health professional migration governance in Nigeria and determine to what extent it is human rights-based), our survey elicited a broad range of perspectives across 38 core items. To derive those unobservable variables that represent the scope of SHW migration governance in Nigeria, we conducted an exploratory factor analysis (EFA) using maximum likelihood factor analysis and Oblimin rotation [46, 47]. We have included details of the statistical methods in a S1 File.

We also conducted a mixed deductive-inductive analysis of the qualitative interviews. First, an audio record of the interviews was transcribed by a professional. KY cross-checked the audio recordings with the interview transcripts to confirm their accuracy and transferred these transcripts into NVivo 12, a computer assisted qualitative data analysis software (CAQDAS). For the deductive analysis, KY designed a codebook using an institutional governance framework for health systems as described by Abimbola (2017) [12] and discussed this with the co-authors. For this study, rules referred to what the participants understood to be a written regulation or law (formal rule), or a shared understanding of what or how things work (norms or informal rules) in response to three broad areas: the actual migratory process, mitigating impact of SHW migration on health services, and factors related to SHW migration intentions.

We coded information on processes, prevalent attitudes or values as factors that influence implementation of various SHW migration governance rules specific for health services or SHW migration. We considered outcomes as attributes that defined what occurred directly or indirectly from the application of the rules. This broad skeletal framework captured governance across three levels–operational (the SHW as an individual who is the primary target of rules from the other levels), collective (groups of stakeholders that are placed between the SHWs and actors at the constitutional level, interacting to either monitor or change the rules or its perceptions), constitutional (the highest level where rules that influence the other two levels are made). In our application of these codes, we remained flexible enough to allow for an inductive derivation of new codes and categories.

KY and RJ then tested and refined the initial codebook after applying it on three transcripts. The rest of the analysis was carried out by KY and followed the approach for thematic analysis as described by Braun (2006) that is, coding of the data (using the revised code book), theme development, reviewing, defining of the themes, and producing the report [48, 49]. To ensure confirmability of the findings, KY kept an audit trail of how themes were formed. Respondent validation was done by sending the coded transcripts to each participant so they could ensure that an accurate interpretation of their statements was captured. Peer debriefing was done by sharing the description of our methods and a sample of coded transcripts with other researchers who were not part of the team. We then combined our findings from the survey with the thematic index from the qualitative study to form a framework that describes SHW migration governance in Nigeria.

For the second objective (which is to determine the relationship between items in the governance framework and migration intentions of health professionals in Nigeria), we conducted univariate and multivariate logistic regression analyses to examine the relationship between the demographic variables, governance items and migration intentions. The dependent variable was migration intentions of the participants comprising of an intention to migrate or to

stay, the independent variables comprised of median scores, proportions, and percentages for the derived SHW migration governance items and demographic characteristics of the study participants. R statistical software was used for all the quantitative analyses [50].

### Ethical clearance

We received ethical approval for this study from the Institutional Research Ethical Committee (IREC) of the Jos University Teaching Hospital on the 17th of July 2020 (reference JUTH/DCSIIREC/127 /XXXl/2249) and from the Health Research Ethics Committee (HREC) of the University of New South Wales on the 7th of August 2020 (reference HC200482). We also obtained written informed consent from all participants prior to data collection.

## Results

### Description of study participants

**Survey.** A total of 372 entries were recorded for the survey. This reduced to 271 valid entries after removing duplicate entries and participants who did not respond to any of the survey questions beyond providing consent for the study. The median age (IQR) of the survey participants was 36 (9) years, median number (IQR) of dependents 4 (2), majority were male (59%, 160/271), and the median (IQR) duration of practice following qualification was 11 (8).

SHWs who had intentions to stay were older (40 versus 35 years) and had a longer duration of practice (13 versus 10 years). Majority (81%, 22/27) of those who practiced in the rural areas had intentions to migrate compared to 55% (133/244) of those who resided in urban areas. Nursing was the profession with the highest proportion of health workers who had intentions to migrate (72%, 49/68). Compared to those who had further training after their initial degree (55%, 135/247), most SHWs who had no further training (83%, 20/24) had intentions to migrate. Further details have been provided in Table 1.

**Semi-structured interviews.** Twenty-two SHWs participated in the semi-structured interviews. This includes six nurses, 10 medical doctors, and six pharmacists. Out of these, eight were females, and 14 were males. All but one SHW practiced in the urban and northern parts of Nigeria with a mean (SD), age of 38.2(7.1) years. Their mean (SD) duration of practice was 11.7 (6.5) years. There was an equal proportion of participants who intended to stay and those who intended to migrate.

**The scope of health professional migration governance in Nigeria—Survey findings.** Following an exploratory factor analysis of the survey findings, we derived eight factors that represent the scope of SHW migration governance in Nigeria: (i) government's efforts towards political and economic stability, (ii) collaborative approaches to SHW migration governance involving the government and other stakeholders within and outside the country, (iii) efforts by civil society organisations and health professional associations that promote awareness on SHW shortages, and promote efforts aimed at discouraging SHW migration, (iv) health workforce policies on recruiting and maintaining an adequate number of SHWs, as well as those encouraging SHWs that have migrated to invest back in Nigeria, (v) governance mechanisms that promote human rights norms/laws (including the right to health, right to fair wages and a good quality of life, and the right to migrate), (vi) efforts by patients and community groups to support SHWs as they remain and practice within Nigeria, (vii) items that reflect SHW's perceived utility of remaining in Nigeria, and (viii) governance outcomes that reflect participants' satisfaction with working conditions and remuneration. Further details about the factor structure matrix, factor correlation matrix, variance explained by each factor and labelling for each factor have been provided in the S1–S5 Files.

**Table 1. Description of the study participants and SHW migration governance items.**

| Characteristic | Overall, N = 271[1] | Intend to stay N = 116[1] | Intend to migrate N = 155[1] |
|---|---|---|---|
| **Age** | 36 (33, 42) | 40 (35, 44) | 35 (32, 40) |
| **Number of dependents** | 4.00 (3.00, 5.00) | 4.00 (3.00, 6.00) | 4.00 (3.00, 5.00) |
| **Duration of practice** | 11 (7, 15) | 13 (10, 17) | 10 (6, 13) |
| **Region** | | | |
| North-Central | 125 (100%) | 50 (40%) | 75 (60%) |
| North-East | 18 (100%) | 9 (50%) | 9 (50%) |
| North-West | 21 (100%) | 8 (38%) | 13 (62%) |
| South-East | 7 (100%) | 4 (57%) | 3 (43%) |
| South-South | 18 (100%) | 6 (33%) | 12 (67%) |
| South-West | 82 (100%) | 39 (48%) | 43 (52%) |
| **Work setting** | | | |
| Rural | 27 (100%) | 5 (19%) | 22 (81%) |
| Urban | 244 (100%) | 111 (45%) | 133 (55%) |
| **Further training after initial qualification** | | | |
| No | 24 (100%) | 4 (17%) | 20 (83%) |
| Yes | 247 (100%) | 112 (45%) | 135 (55%) |
| **Gender** | | | |
| Female | 111 (100%) | 43 (39%) | 68 (61%) |
| Male | 160 (100%) | 73 (46%) | 87 (54%) |
| **Profession** | | | |
| Dentist | 14 (100%) | 10 (71%) | 4 (29%) |
| Medical Doctor | 132 (100%) | 60 (45%) | 72 (55%) |
| Nurse & Midwives | 68 (100%) | 19 (28%) | 49 (72%) |
| Pharmacist | 57 (100%) | 27 (47%) | 30 (53%) |
| **Governance items** | | | |
| Factor 1 | 1.67 (1.00, 2.33) | 2.00 (1.33, 2.33) | 1.67 (1.00, 2.33) |
| Factor 2 | 2.40 (2.00, 3.00) | 2.40 (2.00, 2.80) | 2.40 (1.80, 3.00) |
| Factor 3 | 3.00 (2.25, 3.50) | 2.75 (2.50, 3.50) | 3.00 (2.25, 3.50) |
| Factor 4 | 2.00 (1.50, 2.33) | 2.00 (1.63, 2.50) | 1.83 (1.50, 2.33) |
| Factor 5 | 2.25 (1.75, 3.00) | 2.50 (1.75, 3.00) | 2.00 (1.50, 3.00) |
| Factor 6 | 3.50 (2.50, 4.00) | 3.50 (3.00, 4.00) | 3.50 (2.50, 4.00) |
| Factor 7 | 2.00 (1.75, 2.50) | 2.25 (2.00, 2.75) | 2.00 (1.50, 2.25) |
| Factor 8 | 2.00 (1.67, 3.00) | 2.33 (2.00, 3.33) | 2.00 (1.33, 2.67) |

*Descriptive statistics of the governance items.* Factor 6 was the governance item with the highest median score of 3.5 suggesting a favourable view among all the participants about patient/community group support for SHWs as they provide health services. The item with the lowest median score was Factor 1, perceptions of the government's efforts toward ensuring political, and economic stability. Even though the overall scores were low, those who had intentions to stay had slightly higher scores for Factor 1, Factor 4 (maintaining the required health workforce), Factor 5 (perceptions about an overall commitment to human rights norms in Nigeria, Factor 7 (perceived gains from remaining at home, and Factor 8 (satisfaction with governments' efforts towards improving SHWs' working conditions and remunerations). Details have been provided in Table 1. The Cronbach alpha scores for each of the factors ranged from 0.53 to 0.83, representing an acceptable scale reliability.

*Relationship between demographic variables, derived governance items and SHW migration intentions–survey findings.* From the univariate analysis, an increase in the odds of an intention

**Table 2. Univariate analysis involving SHWs who wish to migrate versus those who wish to stay.**

| Term | [1]OR | std.error | statistic | df | p.value | Lower C.I | Upper C.I |
|---|---|---|---|---|---|---|---|
| (Intercept) | 1.51 | 0.09 | 4.73 | 1350.87 | 0.00 | 1.27 | 1.79 |
| Gender: Male | 0.82 | 0.11 | -1.82 | 1350.87 | 0.07 | 0.65 | 1.02 |
| **Age** | **0.93** | **0.01** | **-8.95** | **1350.87** | **0.00** | **0.92** | **0.95** |
| Number of dependents | 0.97 | 0.01 | -2.13 | 1350.87 | 0.03 | 0.95 | 1.00 |
| **Profession | Medical Doctor** | **2.89** | **0.27** | **3.93** | **1348.87** | **0.00** | **1.70** | **4.92** |
| **Profession | Nurse** | **6.43** | **0.29** | **6.49** | **1348.87** | **0.00** | **3.66** | **11.29** |
| **Profession | Pharmacist** | **2.73** | **0.28** | **3.53** | **1348.87** | **0.00** | **1.56** | **4.78** |
| **Duration of practice** | **0.92** | **0.01** | **-9.87** | **1350.87** | **0.00** | **0.90** | **0.93** |
| **Further training | Yes** | **0.25** | **0.25** | **-5.71** | **1350.87** | **0.00** | **0.15** | **0.40** |
| Region | North-East | 0.70 | 0.23 | -1.60 | 1346.87 | 0.11 | 0.45 | 1.08 |
| Region | North-West | 1.33 | 0.22 | 1.28 | 1346.87 | 0.20 | 0.86 | 2.07 |
| Region | South-East | 0.52 | 0.35 | -1.85 | 1346.87 | 0.06 | 0.26 | 1.04 |
| Region | South-South | 1.44 | 0.24 | 1.54 | 1346.87 | 0.12 | 0.90 | 2.29 |
| Region | South-West | 0.78 | 0.13 | -1.94 | 1346.87 | 0.05 | 0.61 | 1.00 |
| **Work setting | Urban** | **0.23** | **0.25** | **-5.90** | **1350.87** | **0.00** | **0.14** | **0.37** |
| **Factor1** | **0.74** | **0.07** | **-4.43** | **1350.87** | **0.00** | **0.65** | **0.85** |
| **Factor 2** | **1.20** | **0.08** | **2.30** | **1350.87** | **0.02** | **1.03** | **1.40** |
| Factor 3 | 0.93 | 0.07 | -1.01 | 1350.87 | 0.31 | 0.82 | 1.07 |
| **Factor 4** | **0.61** | **0.09** | **-5.62** | **1350.87** | **0.00** | **0.51** | **0.72** |
| **Factor 5** | **0.80** | **0.06** | **-3.59** | **1350.87** | **0.00** | **0.71** | **0.90** |
| **Factor 6** | **0.73** | **0.06** | **-5.65** | **1350.87** | **0.00** | **0.66** | **0.82** |
| **Factor 7** | **0.13** | **0.14** | **-15.05** | **1350.87** | **0.00** | **0.10** | **0.17** |
| **Factor 8** | **0.67** | **0.06** | **-6.64** | **1350.87** | **0.00** | **0.59** | **0.75** |

**Key:** [1]OR = Unadjusted Odds Ratio, **CI** = Confidence Interval, **shaded items are statistically significant**; **SHWs**- skilled health workers, **Factor 1**—Government's efforts towards political, and economic stability, **Factor 2**—Cooperative approach to SHW migration governance, **Factor 3**—Efforts by non-state actors, **Factor 4** – Specific health workforce policies, **Factor 5**—Commitment to human rights broadly, and the RTH specifically, **Factor 6**—Efforts by patients and community groups to support SHWs, **Factor 7** SHWs' perceived benefit of remaining in Nigeria and **Factor 8** –SHW's satisfaction with government's efforts towards improving working conditions and remuneration for SHWs

to migrate was associated with belonging to the nursing profession and perceptions about collective action towards SHW migration governance. Conversely, an increase in age and duration of practice, acquiring further training after the initial medical qualification, and practicing in an urban setting was associated with a decrease in the odds of an intention to migrate. Similarly, Factor 1 (perception of government's effort towards ensuring economic and political stability), Factor 4 (relevant health workforce policies), Factor 5 commitment to human rights norms/laws), Factor 7 (SHW's perception of staying in Nigeria compared to migrating), and Factor 8 (satisfaction with government's efforts towards improving SHW working conditions and remuneration) were associated with a decrease in the odds of an intention to migrate Table 2.

For the multivariate analysis, when both demographic and governance items were included in a model, being a nurse, an interaction between duration of practice and receiving further training, and perceptions about collaborative approaches in the governance of SHW migration (Factor 2) were associated with an increase in the odds of an intention to migrate. Conversely, duration of practice, receiving further training in Nigeria, and living in an urban setting; positive perceptions about health workforce policies (Factors 4), support from patient/community groups (Factor 6), staying in Nigeria compared to migrating (Factor 7), and satisfaction with

**Table 3. Multivariate analysis involving SHWs who wish to stay (reference) versus those who wish to migrate from Nigeria.**

| | Demographic variables only | | | Governance items only | | | All items | | |
|---|---|---|---|---|---|---|---|---|---|
| | OR | Lower C.I | Upper C.I | OR | Lower C.I | Upper C.I | OR | Lower C.I | Upper C.I |
| Gender: Male | 0.85 | 0.65 | 1.11 | | | | 0.89 | 0.65 | 1.23 |
| Age | 1.04 | 0.99 | 1.09 | | | | 1.04 | 0.99 | 1.10 |
| Number of dependents | 0.99 | 0.96 | 1.02 | | | | 0.99 | 0.95 | 1.03 |
| **Profession \| Medical Doctor** | **3.98** | **2.07** | **7.64** | | | | **2.79** | **1.40** | **5.54** |
| **Profession \| Nurse** | **7.48** | **3.81** | **14.67** | | | | **5.35** | **2.67** | **10.70** |
| **Profession \| Pharmacist** | **3.45** | **1.76** | **6.80** | | | | **2.36** | **1.18** | **4.73** |
| **Duration of practice** | **0.33** | **0.20** | **0.55** | | | | **0.91** | **0.86** | **0.97** |
| **Further training \| Yes** | **1.08** | **0.98** | **1.19** | | | | **0.27** | **0.15** | **0.49** |
| **Duration of practice * Further training\|Yes** | **0.14** | **0.05** | **0.39** | | | | **1.13** | **1.01** | **1.26** |
| Region \| North-East | **0.83** | **0.75** | **0.93** | | | | **0.50** | **0.29** | **0.87** |
| Region \| North-West | **0.48** | **0.29** | **0.79** | | | | 1.43 | 0.80 | 2.55 |
| Region \| South-East | 1.09 | 0.68 | 1.75 | | | | 1.01 | 0.44 | 2.33 |
| Region \| South-South | 0.56 | 0.27 | 1.18 | | | | 1.08 | 0.59 | 1.99 |
| Region \| South-West | 1.41 | 0.85 | 2.34 | | | | 0.87 | 0.62 | 1.21 |
| **Work setting \| Urban** | **0.83** | **0.62** | **1.11** | | | | **0.50** | **0.28** | **0.89** |
| Factor1 | | | | 1.19 | 0.97 | 1.46 | 1.24 | 0.99 | 1.56 |
| **Factor 2** | | | | **2.02** | **1.61** | **2.54** | **1.94** | **1.51** | **2.49** |
| Factor 3 | | | | 1.00 | 0.84 | 1.19 | 0.96 | 0.79 | 1.16 |
| **Factor 4** | | | | **0.74** | **0.57** | **0.97** | **0.73** | **0.54** | **0.98** |
| Factor 5 | | | | **1.28** | **1.06** | **1.55** | 1.16 | 0.94 | 1.43 |
| **Factor 6** | | | | **0.75** | **0.65** | **0.86** | **0.71** | **0.60** | **0.85** |
| **Factor 7** | | | | **0.11** | **0.08** | **0.15** | **0.13** | **0.09** | **0.17** |
| **Factor 8** | | | | **0.67** | **0.57** | **0.78** | **0.76** | **0.64** | **0.91** |

**Key: OR** = adjusted Odds Ratio, **CI** = Confidence Interval; * interaction between items, **shaded items are statistically significant**; **SHWs**- skilled health workers, **Factor 1**—Government's efforts towards political, and economic stability, **Factor 2**—Cooperative approach to SHW migration governance, **Factor 3**—Efforts by non-state actors, **Factor 4** –Specific health workforce policies, **Factor 5**—Commitment to human rights broadly, including the right to health, **Factor 6**—Efforts by patients and community groups to support SHWs, **Factor 7** SHWs' perceived benefit of remaining in Nigeria and **Factor 8** –SHW's satisfaction with government's efforts towards improving working conditions and remuneration for SHWs

remuneration/working conditions were associated with an intention to stay in the country. Further details are available in Table 3.

**Scope of SHW migration governance in Nigeria–Qualitative findings.** In this section we identified themes and exemplar quotes that reflect the participant's understanding of governing rules/norms for SHW migration in Nigeria, factors influencing accountability to these rules, and their outcomes across the constitutional, collective, and operational levels. The thematic index of the qualitative findings can be found in the S6 File.

**Constitutional level: Governing rules/norms and factors affecting accountability.** *Funding health services.* There are formal rules on how the government should fund health services in Nigeria. According to one of the participants, *". . .the Basic Healthcare Provision Act. . .stipulates that at least 1% of the National Consolidated Fund be set aside. In the disbursement of this fund . . ., 45% will go to the . . .National Primary Healthcare Agency to provide drugs, services, . . . and about 5% will go into emergency [health services]"* (P22). Even though there is a formal rule that should guide funding of health services, another participant thought that *". . .there [was] no [political] will, enforcement, or execution by the government. . .***(P7)** There were also *". . .issues with continuity and maintenance so even though a particular*

*government. . . starts a program, most of the time when that government leaves, . . . things just fall apart"* (P17). Other factors influencing the government's accountability to norms/rules on funding include their perception of the right to health. For instance, one participant thought the *". . .government . . .do[es] not care if Nigerians have [the] right to health."* (P17). Another explained that *". . .when [the government] thinks about securing right to health, [they] focus [only] on. . .hospitals,. . . but really health is much bigger than that. . ."* (P23)

*Managing the skilled health workforce.* Concerning how the skilled health workforce is managed in Nigeria, one participant explained that *"[The] Nigerian government subsidises training for healthcare workers [. . .] believe[ing] that when they finish, they will work in the country"* **(P22).** In addition, all SHWs in the public health sector [including those in training] are paid by the government. As one participant mentioned, *". . .public health institutions in Nigeria . . .get their staff free of charge. The government. . .pays everybody, so all the [institutions] have to do is manage whatever internally generated revenue [they] get"* (P17).

Even though the government subsidises training for SHWs, there are still shortages in the skilled health workforce mainly because the government is not recruiting enough. As one participant pointed out, there are lots of *". . .medical doctors, nursing officers, pharmacists* [without jobs]" (P14).

For SHW's already employed by the government, the participants noted that salary adjustments only happened following strikes and protests about poor working conditions and remunerations. One participant attest to this, recalling that he *". . . joined the service when . . .salary adjustments [had just started]"* (P2). While salary adjustments do occur, a participant criticised the government for thinking that *". . .the [SHW working in the] tertiary hospital. . .deserves to be paid more"* than those working at the secondary health centres (P12). Altogether, participants thought that the government paid little attention to the health system, and that the remuneration package for Nigerian SHWs was far less than what SHWs should receive. According to this participant, *". . . a comparative analysis of remuneration in African countries [revealed] that over forty countries in Africa pay their doctors and other healthcare workers better than Nigeria"* (P22).

*Skilled health worker migration.* Participants thought that the government was not concerned about SHW migration and were not aware of collaboration with stakeholders within Nigeria, or bilateral agreements on recruitment of SHWs. One participant remarked that the government *". . .wants [health workers] to go, and then send money back, and then they can use that money they send back to bail the nation"* (P16). Similarly, another participant noted that the government's actions can be seen as them saying *". . .we do not want to increase [health workers] salary, let's just allow them go, so that they will not be disturbing us with strike and other things [. . .]"* (P25).

Some participants were aware of attempts by state governments to regulate SHW migration through bonding schemes. As one participant attested, *"I know. . .of a few states in the north that the [state] government sponsored their citizens to go and read medicine and nursing at the university and required them to come and put back a number of years of service, before they are allowed to even transfer to a different state within, or outside Nigeria."* (P23).

However, most of the participants mentioned that formal rules guiding the retention of SHWs was lacking at the national level. As one participant remarked, *"I do not think there are any formal rules that I know of in Nigeria that govern the way stakeholders are addressing or not addressing health worker migration. So, there is no law or framework, . . . I am not aware of any document that governs it."* (P23).

Referring to international initiatives for regulating SHW migration, one participant shared that the WHO Code has not been mentioned *". . .in any formal setting [*she had attended*]. It is not something that has been discussed in hospitals, [nor] within professional health associations.*

[She was not aware] *that it is something that Nigeria formally recognises* (P23). Another participant heard about the WHO Code when he was applying for a job in the UK. The recruiting agency he contacted advised that there was a way to get round the Code, that is, by *". . .apply [ing] directly [to the hospital], which means [the job] had not been advertised, but it is the individual that applies directly [to the hospital that gets it]"* (P14).

**Constitutional level: Governance outcomes.**   When asked about the impact of governance at the constitutional level, the participants described a lack of universal health coverage and realisation of the right to health. One participant mentioned that many health facilities do *". . .not have the right drugs to use, or* [must function where] *there is insecurity. [Hence, people] may have the right* [to health] *but cannot exercise it"* (P10). Another remarked that *". . .only about 5% of Nigerians are covered by the national health insurance despite the fact that it has operated for more than . . . 15 years now"* (P4).

**Collective level: Governing rules/norms and factors affecting accountability.**   *Social and financial support for SHW migration.* In the absence of formal regulations on SHW migration, participants shared their experiences about how family members and private financiers influence SHWs' search for better career prospects abroad. One participant recalled an experience when a colleague's wife learnt he had turned down an opportunity to work abroad, *". . .she reported him to his family. His father called him, his mother called him, and everybody was shouting at him [to] leave" (*P17). In contrast, some participants observed that colleagues who had lots of family responsibilities and were advanced in age thought that they were *". . . no longer in a good position to leave the country. . ." (*P26). Another participant observed that when SHWs lack financial resources to support their migration, *". . .there are some people that can [take the] responsibility for that. They [will] pay for everything and at the end of [the migration process] . . .based on [an] agreement, . . . take [a] percentage of"* the SHW's salary at their new job overseas (P2).

*Advocacy and engagement by non-state actors.* The participants thought that only non-state actors were advocating for better policies to address the impact of SHW migration in Nigeria. As described by one participant, *". . .most of the efforts [in response to SHW migration] were from . . . civil society organisations and non-governmental organisations. . ."* (P22). The participants did not think other members of the society played a particular role in the governance of SHW migration. One participant blamed the absence of societal inputs on the lack of *". . .civic sense of responsibility, community engagement and participation,"* which in turn could be traced to a *". . .general [sense] of disregard of people's rights and how [the government] has used citizens [to gain] political currency." (*P3). Another participant thought poor societal participation was because many Nigerians *". . .do not even know they have the right [to health] . . .*(P14). Yet another blamed it on how SHWs are perceived, remarking that *". . . the society generally thinks that [SHWs] are pompous [and] do not care what happens to [the masses]"* (P13).

*Oversight by health administrators/regulatory bodies.* Some participants were convinced that hospital administrators *". . . have a role to play in the [migration process of SHWs]."* Before they travel, SHWs are required to get the approval of their hospital administrators and sign an agreement stating when they will return to work (P2). In addition, they thought health practitioner regulatory bodies were not doing enough to regulate SHW migration. One participant though this lack of input was due to *". . . a laissez faire attitude towards the migration of health workers. . . .They [the regulatory bodies] show less concern. . ."* if a health worker migrates or stays (P26).

Regarding working conditions within health facilities, participant blamed hospital administrators for not prioritising health worker safety. One participant described this as follows, *". . .we [. . .] found ourselves in a situation where the personal protective equipment we were supposed to be provided with, were not [. . .] provided."* (P14). The participants also thought that

the hospital administrators were responsible for the workforce shortages in the hospitals. One participant thought that *". . .if the [hospital administrators] want to recruit, they will recruit. It is because they do not want to recruit [that SHW shortages persists]"* (P12). A lack of professional and human rights norms explains accountability at this level, as one participant remarked that leaders in the health sector *". . .do not have [a] heart for the [health] profession, most of them do not really [care for] humanity. . ."* (P26).

*Health worker resilience, interactions, and advocacy.* Participants thought that it was common to see SHWs as a collective group adapt to the challenging work conditions in the Nigerian health system and attributed this to existing professional norms. As one participant stated, *"I think we get to improvise a lot in this environment, and I think it hones one's problem-solving ability. . ."* (P22). By *". . .conducting community campaigns, [health] education, . . .screening exercises for different diseases."* (P17), SHWs were making efforts towards improving their job satisfaction in a challenging work environment. Though a participant had concerns about *". . .inter and intra professional rivalry, . . ."* (P3), another participant described pleasant experiences at work, and attributed this to a mutual *". . .recognition of [professional] roles, [and] teamwork. . . ensur[ing] that ultimately, the patient benefits"* (P23).

Concerning interactions among SHWs, one participant remarked that *". . .a network of health professionals talking to each other [influenced SHW] migration [intentions]"* (P23). In addition, they noted a moral dilemma concerning migration among SHWs. One participant described this dilemma as follows, *". . . [Health workers struggle to decide. They might be thinking that], this is a country that has trained me and has subsidised my training, should I just leave and go?"* (P22)

Another participant mentioned SHWs' advocacy efforts, commenting that *". . .the leadership [of the Nigerian Medical Association] advocates . . . for government to deploy [resources towards] improving healthcare delivery. . ."* (P4). The participants thought SHW groups did not collaborate much with civil society organisations, and by themselves had done little to address the increase in health worker migration. In this regard, one participant remarked that *". . .in theory, "most of the practitioners in the health sector subscribe to [the right to health] and value it. But in actual sense, [they are not committed] to [its] implementation"* (P4). Another mentioned that *". . .some of the health workers are not even acquainted with the Health Act. So, when you have a society where people are not educated on certain things, the government [gets] away with anything"* (P25).

**Collective level: Governance outcomes.** The outcomes here were specific for patients and SHWs. One participant described how *". . . patients come to [the hospital] and . . .have to wait for long hours. . .before [they are] attended to"* (P15). Though SHW training is associated with *'. . .some challenges. . ."* (P25), another participant thought that the decision to train in Nigeria led to good academic outcomes and *". . . [changed his] mentality towards life generally"* (P26). Yet another thought that the health workforce has been poorly managed, hence SHWs *". . .only do [routine tasks] . . ."* and are unable to commit to *". . .what will make a difference to the patient. . ."* (P8).

**Operational level: Governing rules/norms and factors affecting accountability.** *Health advocacy and engagement.* Some of the participants mentioned that their role in SHW migration governance involved *". . .influencing [their] peers and creat[ing] awareness [among] patients and colleagues that [the right to health] is a fundamental human right."* (P23). Another participant mentioned *". . .engag[ing] policy makers one-on-one concerning [improving health services] . . .,"* engaging with the media about health worker migration and helping communities *"..understand health as a right, and that the people need to hold government accountable, and demand [for] the right investment in health care"* (P22). While some participants demonstrated commitment to building the health system, others expressed a sense of despair as

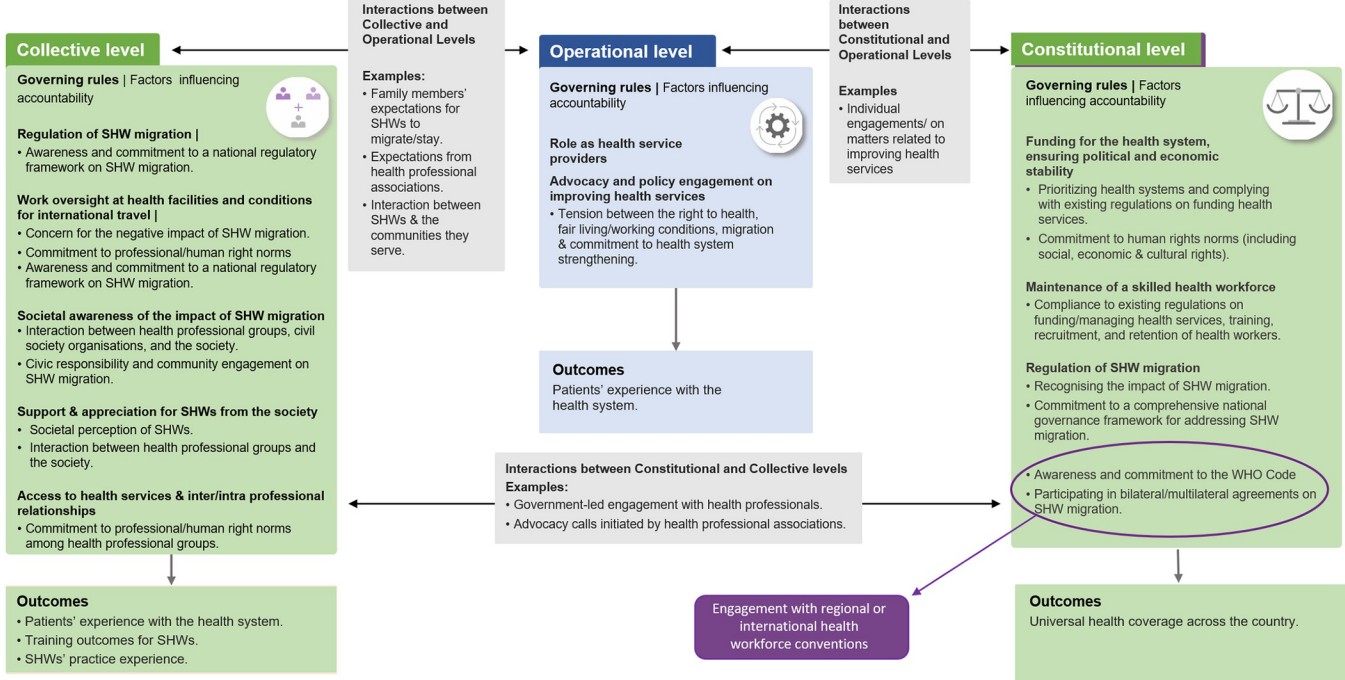

**Fig 2. Governance framework for skilled health worker migration in Nigeria.**

captured by one participant, *"so, whether you drop dead at your place of work or you are in the grave, the health system will not lack [health workers]. Soldier go, soldier come, barrack no dey shake." [a Nigerian proverb that describes how soldiers are dispensable, but the barracks remains] (P12).*

*Health service delivery.* Most of the participants thought it was their responsibility to provide quality care for their patients: *"there is [a] social contract that has been signed between me and that patient, that they have the right to get from me the best care that is available... I owe it to myself, and I owe it to them to do the best that I can do" (P23)*. Even though participants subscribed to the right to health, they also upheld their right to migrate. As one participant mentioned, *"...every human being also has the right to migrate including healthcare workers. If an accountant can migrate, if a lawyer can migrate, why can't [health workers] migrate?" (P20).*

**Operational level: Governance outcomes.** Most of the participants mentioned that they enjoyed their clinical practice. As one participant mentioned, *"though it has been very challenging and tasking, but honestly, I think I have a . . . satisfying experience to date" (P9)*. Another participant attributed their sense of fulfillment to the opportunities she had to improve health outcomes of her patients: *"When you get to treat someone and see the person recover, . . . you are able to make an impact that gives you that satisfaction [. . .]" (P17)*

**Triangulating the findings.** Most of the quantitative and qualitative findings were in alignment (See S7 File) and have been merged into a framework (Fig 2) which offers a conceptual approach for describing and analysing the governance of SHW migration in Nigeria.

## Discussion

We described the scope of SHW migration governance in Nigeria across three levels: constitutional, collective, and operational. At the constitutional level, the participants mentioned governing rules/norms that address funding of health services, domestic health workforce issues

(including work conditions, salaries, training, and retention of SHWs), political, economic, and social stability. Even though they recognised some efforts by the Nigerian government towards implementing bonding schemes on SHWs, increasing SHW remuneration and subsidising their training, they observed as unsatisfactory, their performance on implementing health workforce policies, ensuring economic stability, and regulating active recruitment of SHWs. At the collective level, participants described social and financial norms that support/constrain SHW migration; interactions and advocacy on SHW shortages by actors within and outside the health system; expressions of gratitude and support to SHWs by patient/community groups; work oversight within health facilities, and efforts aimed at improving access to health services. In addition, the participants thought there was a lack of civic responsibility and community engagement at this level, an attitude of indifference concerning the impact of SHW migration, and a lack of collaboration between health professional groups and civil society organisations.

At the operational level, participants reflected on their duty as health service providers, and their responsibility towards career progression, personal and family wellbeing. To a lesser extent, they described their participation in policy advocacy and engagement on issues related to SHW migration. Even though they acknowledged that the right to health influenced their approach to health service delivery, they admitted that more could be done regarding advocating for a human rights-based approach to SHW migration governance. When derived governance items were considered in a regression model, a commitment to human rights norms did not show a statistically significant association with migration intentions. Similarly, a perception of increase in collective action towards SHW migration regulation was associated with an increase in the odds of an intention to migrate.

Rights-based governance is driven by an understanding of International Human Rights Law (IHRL) which recognises freedoms/entitlements of individuals and the responsibilities of states to uphold them. It also recognises a social order where countries cooperate to solve complex, wicked problems [39, 41, 43, 51–53]. Whereas our finding suggests a limited uptake of rights-based approaches for handling SHW migration in Nigeria, there are studies that have highlighted its relevance for LMICs. A good example in this respect is the Managed Migration Program (MMP) in the Caribbean where SHW's respect for the right to health and their right to migrate influenced the approach to SHW migration governance [54, 55]. The MMP was set up to promote collaboration between state and non-state actors concerned with training, recruitment, and retention of health workers [54, 55]. Scholars have ascribed the success of this strategy to a clarity of goals, the leadership and dedication of SHWs and other non-state actors who subscribed to the idea, and political commitment from the governments in these countries; features that our participants thought were lacking in the Nigerian context [54, 55]. The MMP exemplifies how health advocates can promote collective action among state and non-state actors using human rights norms.

A less visible presence/influence of human rights norms within the governance framework for SHW migration as suggested by our study, is not peculiar to Nigeria. An earlier study in selected African countries also revealed a lack of awareness of legal instruments useful for promoting human rights among health workers and the general population. This lack of awareness happened even though these countries were signatories to many human rights treaties [56]. There are however good examples of how human rights norms have been used to promote public health. For example, as democracy became established in South Africa, civil society actors relied on human rights principles as they took legal action against the government, enabling a political discourse on improving access to HIV medicines [57]. Such legal invocation of the right to health in South Africa can be traced back to the political and social struggles

they have experienced as a people. These struggles have entrenched the political and legal entity of human rights in South Africa alongside its ethical/moral entity [58].

Promoting the right to health often reveals a tension with SHWs right to migrate. The WHO Code seeks to address this tension by promoting ethical recruitment of SHWs, bilateral/multilateral agreement on health workforce migration, and domestic efforts towards achieving a sustainable health workforce. Drawing from experiences in Sudan, the WHO Code can help trigger public discourse on SHW migration and bilateral agreements on its regulation in Nigeria [59]. However, our study suggests a low awareness of the WHO Code or its role in the governance of SHW migration in Nigeria, consistent with findings from a study exploring its relevance in eastern and southern Africa [60] Even if poor engagement with the WHO Code can be explained by state actors' disagreement with its relevance in an African context [60], the absence of civic engagement on SHW migration and its impact on population health suggests a more fundamental issue regarding how this issue is perceived in Nigeria. Previous studies have shown that social mobilisation for improving population health in Nigeria is possible [6, 61, 62] Since our findings offer little evidence in support of social mobilisation for improved SHW migration governance, it may reflect a lack of consensus on how SHW migration governance offers public value in Nigeria. Beyond responding to push and pull factors, scholars have described migration behaviour as an expression of liberty—the freedom to choose where, and how to live [63]. Our participants may have envisioned increased constraints on their personal liberties, if they are expected to support collective action to regulate SHW migration (Factor 3) and commit to promoting human rights norms (Factor 5).

Owing to its importance, there are concerns among scholars about the limited role of human rights norms (especially the right to health) for influencing SHW migration governance [64]. The right to health has not been frequently used to describe or address the impact of SHW migration despite its role in addressing equitable access to medical treatments [64–66]. If like the MMP program in the Caribbean, SHWs play a key role in promoting rights'-based approaches for migration governance, our findings suggest that they have not yet been able to navigate the conflicts around promoting the right to health, their right to free movement, fair remuneration, living and working conditions [65, 66]. If the normative value of human rights for SHW migration governance is constrained by its perceived limitation of individual/private value, then its uptake in the Nigerian political and legal system may also be constrained. Like other LMICs, a political/legal invocation of the right to health in Nigeria is hindered by a lack of relevant resources (financial, operational etc) for its realisation, and absence of accountable mechanisms (including legal institutions and competent practitioners) [66, 67]. Despite these constraints, a previous study revealed that the right to health might still offer utility for addressing SHW shortages in LMICs when used to negotiate workforce agreements and engage stakeholders with competing interests [68]. Such stakeholder engagements can lead to innovative application of the WHO Code and other initiatives (including a whole of government/society approach) useful for mitigating the negative impact of SHW migration [54, 55, 68].

## Strengths and limitations

In adopting an institutional approach, we affirm existing recommendations for a whole of government/society approach for handling SHW migration [69] and the implications this has for setting up relevant governance structures ("hardware"). Our approach also recognises the need for relational ("soft-ware") aspects that should be considered in understanding the scope of SHW migration governance [70, 71].

By eliciting the perceptions of our participants, we align with an interpretivist approach to governance; one that explains governance through actions, explores belief and meaning for an in-depth understanding of these actions, recognises governance as including top-down and bottom-up practices, and explores its contingent nature (that is, governance as having no definite causal path) [12, 72]. Underpinning these actions is an interplay of power. Hence, we recognise that a post-structuralist governmentality of health worker migration offers an alternative analytical framework [73]. However, in Nigeria (as in many other LMICs), there is a tendency for state actors to under govern. Hence, our preference for governance as an analytical tool, specifically, a structure-relational approach [12, 74].

The subjective judgements of our participants are relevant to a specific context, and thus, cannot be generalised. However, we do not consider this to be a limitation to the usefulness of our framework. SHWs are the primary targets of any formal policy/regulation on migration. Their perception and response to existing rules (or a lack of rules) drives migration behaviour [75, 76]. In addition, their reasoning can be influenced by interactions (or lack thereof) with other relevant stakeholders. As with all complex, wicked problems, a frame of reference is needed to guide a country's response to SHW migration [77]. Beyond an understanding of the scope of skilled health migration governance in Nigeria, our framework provides multiple entry points and interconnections useful for understanding how SHW migration is handled in under governed settings. It thus offers insights for designing future policy interventions and evaluating their utility for achieving health workforce sustainability in Nigeria and other national settings facing similar challenges.

## Implications for stakeholders

SHW migration governance remains a topical issue especially as more HICs have increased recruitment of foreign trained health workers in response to the COVID pandemic [78]. Adopting a rights-based approach will require skilful navigation of the tension between the right to health, SHWs' right to free movement, and their right to optimal living/working conditions. Protecting and fulfilling these rights cannot be done exclusively by a national government. It will require collective action involving its subnational units, non-state actors and foreign governments. Hence, our framework provides a way to conceptualise the layers of SHW migration governance in Nigeria (or other countries with similar challenges) and highlights opportunities for improving interaction between relevant stakeholders.

A collaborative approach for improving SHW migration governance will depend on the existing social capital between state and non-state actors. Our framework provides a guide for community groups, health professional associations, civil society organisations to understand their role in building the required social capital. While it will be difficult to address competing interests around SHW migration in Nigeria, human rights norms can complement existing norms of solidarity in defining the principles of engagement, creating a system of accountability, and shaping motivations for collaboration [79]. In this regard, our framework offers scholars and health advocates a tool for further institutional analysis of SHW migration governance in Nigeria and similar settings.

## Conclusion

While there are governing rules/norms that should define state actors' response to SHW migration in Nigeria and the realisation of health equity, our study suggests poor accountability to these rules/norms and the perception that state actors have not recognised the impact of SHW migration on the health indices of the country, do not engage frequently with relevant

stakeholders, and have not shown commitment to human right norms. These norms include the right to health and SHWs' right to fair remuneration, living and working conditions.

In describing a governance vacuum at the constitutional level, our participants perceived hospital administrators and health professional regulatory agencies as less inclined to exercise their oversight functions, leaving private recruiters and financial agencies that support SHW migration to operate freely. They thought health professional groups, journalists and civil society actors were drawing attention to the existing governance vacuum, while health professional groups were making efforts towards addressing access to health services. However, they considered these attempts to be constrained by a focus on survival, poor commitment to human rights norms, a lack of solidarity, and collaboration at the collective level of governance.

While SHWs recognise their role as health service providers, and a few engage with other stakeholders on issues related to SHW migration, there is little consensus on their roles as activists or policy entrepreneurs, or their role in health system strengthening. This reflects their perception of a low level of interaction with the government and the communities they serve, a lack of assurance that their inputs will make a difference; the tension between their recognition of the right to health, their right to fair remuneration, living and working conditions.

Adopting a rights-based approach to SHW migration governance will require that the government recognise the need for a whole of government/society approach, seek to engage more with foreign governments, and invite citizens to hold it accountable to an agreed set of processes and outcomes. It will also require that the media, civil society actors, community groups and SHWs collaborate to create awareness for rights-based approaches, complementing existing norms of solidarity and enabling social mobilisation for improving SHW migration governance.

## Supporting information

**S1 File. Steps taken in conducting an exploratory factor analysis.**
(DOCX)

**S2 File. Factor structure matrix.**
(DOCX)

**S3 File. Factor correlation matrix.**
(DOCX)

**S4 File. Variance explained by an eight-factor model.**
(DOCX)

**S5 File. Labelling for each factor.**
(DOCX)

**S6 File. Thematic index of the qualitative findings.**
(DOCX)

**S7 File. Mapping the quantitative results to the qualitative findings.**
(DOCX)

## Acknowledgments

I acknowledge the contributions of Nancy Briggs (Senior Statistical Consultant, UNSW Research Infrastructure Stats Central), Michelle Lim (Statistical Assistant, UNSW Research Infrastructure Stats Central) and Robert Fletcher (Associate Biostatistician, The George Institute for Global Health) for their guidance on my statistical methods. I also acknowledge Prof

Andrea Durbach (Professor Emeritus, Faculty of Law, University of New South Wales) for her feedback at the conception stages of this study and Dr. Seye Abimbola (Senior Lecturer, School of Public Health, University of Sydney) for his feedback on the qualitative analysis and conceptual framework.

## Author Contributions

**Conceptualization:** Kenneth Yakubu.

**Data curation:** Kenneth Yakubu.

**Formal analysis:** Kenneth Yakubu.

**Investigation:** Kenneth Yakubu.

**Methodology:** Kenneth Yakubu.

**Project administration:** Kenneth Yakubu.

**Software:** Kenneth Yakubu.

**Supervision:** Janani Shanthosh, David Peiris, Rohina Joshi.

**Validation:** Kenneth Yakubu, Janani Shanthosh, Kudus Oluwatoyin Adebayo, David Peiris, Rohina Joshi.

**Writing – original draft:** Kenneth Yakubu.

**Writing – review & editing:** Kenneth Yakubu, Janani Shanthosh, Kudus Oluwatoyin Adebayo, David Peiris, Rohina Joshi.

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
