## [Decision Letter · Decision Letter 0]

22 Jul 2022

PGPH-D-22-00874

Scope of health worker migration governance and its impact on emigration intentions among skilled health workers in Nigeria

Dear Dr. Yakubu,

Thank you for submitting your manuscript to PLOS Global Public Health. After careful consideration, we feel that it has merit but does not fully meet PLOS Global Public Health’s publication criteria as it currently stands. Therefore, we invite you to submit a revised version of the manuscript that addresses the points raised during the review process.

We look forward to receiving your revised manuscript.

Kind regards,

Julia Robinson

Executive Editor

Journal Requirements:

1. Please amend your online Financial Disclosure statement. If you did not receive any funding for this study, please simply state: “The authors received no specific funding for this work.”

2. Please update your online Competing Interests statement. If you have no competing interests to declare, please state: “The authors have declared that no competing interests exist.”

3. We do not publish any copyright or trademark symbols that usually accompany proprietary names, eg (R), (C), or TM  (e.g. next to drug or reagent names). Please remove all instances of trademark/copyright symbols throughout the text, including © (SurveyMonkey©) on page 7.

4. We have noticed that you have uploaded Supporting Information files, but you have not included a list of legends. Please add a full list of legends for your Supporting Information files after the references list.

Additional Editor Comments (if provided):

Reviewers' comments:

Reviewer's Responses to Questions

**Comments to the Author**

1. Does this manuscript meet PLOS Global Public Health’s publication criteria? Is the manuscript technically sound, and do the data support the conclusions? The manuscript must describe methodologically and ethically rigorous research with conclusions that are appropriately drawn based on the data presented.

Reviewer #1: Yes

Reviewer #2: Yes

2. Has the statistical analysis been performed appropriately and rigorously?

Reviewer #1: Yes

Reviewer #2: I don't know

3. Have the authors made all data underlying the findings in their manuscript fully available (please refer to the Data Availability Statement at the start of the manuscript PDF file)?

Reviewer #1: Yes

Reviewer #2: Yes

4. Is the manuscript presented in an intelligible fashion and written in standard English?

Reviewer #1: Yes

Reviewer #2: Yes

5. Review Comments to the Author

Reviewer #1: This is an interesting paper, one that can contribute to the literature on SHW migration. I have a few suggestions to strengthen the paper

1 I appreciate the focus on the rights based migration approach, but you might also want to relate this what some authors call 'whole of government or whole of society' approach, for example see Buchan, J., Campbell, J., Dhillon, I., & Charlesworth, A. (2019). Labour market change and the international mobility of health workers. Health Foundation working paper, 5.

2. I appreciate the multilevel analysis of the data, but you are missing a fourth level, that of the international or global level. You mention the WHO Code, which would be relevant to this scale, but also relevant are bilateral agreements, regional mobility agreements, international conventions that apply (even if they are ignored). You do mention some of these agreements but you have not included this level in your model. It is relevant even if that relevance is due to public and state reluctance to engage with it.

3. Under 'material and methods' the section on sample size needs to be more clearly explained, including providing the actual number of surveys and interviews conducted versus planned, and the relationship of the survey results to the interview script. Also a table might be useful to show the different methods used and how they related to each other (what order they occurred in and how one influenced or not the other). The literature review is also a method, and so more information on how this was conducted and how it informed other methods would be useful.

4. The discussion on the balance between right to health and right to migrate is also key to the WHO Code, which indicates SHW right to migration but also in light of their legal responsibilities. You could discuss the Code in more detail, particularly if you include it as one of the levels of analysis. Also your comments on the Code drawn from the interviews needs to be more contextualized. You need to briefly explain the code and its relevance to Nigeria (Nigeria is not on the safeguard list, but is a member of WHA so must have approved the Code in 2010). Since your focus is on governance the global governance aspect of this phenomena should be considered.

5. You mention on p28 that this approach is a useful tool of analysis to explore SHW in Nigeria, but you should also consider how it can be used in other national settings. You should be more assertive in the contribution your approach can make to the literature on SHW migration.

Reviewer #2: This is article focuses on the governance of health worker migration as perceived by health workers themselves. I am not especially convinced by the methodology and methods used, but I recognise that there are other views in the field and that a post-structuralist 'governmentality' of health worker migration might, for many colleagues, be a viable approach. Nevertheless, the authors frame this in terms of governance, by which I think they probably mean governance, regulation and policy, which is different from governmentality. The paper is well-written and -presented. The data on health worker (e)migration (including return migration) could meaningfully and usefully be updated (presently it stops at 2011 - more than 10 years ago!).

Suggested revisions:

1. Clarify and justify the choice of governance as an analytical device (as opposed to regulation, policy, law and governmentality).

2. Clarify and justify the decision to focus on subjective understandings of individual migrant health workers viz governance and what new perspectives this potentially brings.

3. Update no.s of emigrant Nigerian health health workers working overseas.

4. Clarify whether any of the sample were return migratnts.

5. Update discussion/conclusions to reflect these changes requested by this and the other reviewer.

6. PLOS authors have the option to publish the peer review history of their article (what does this mean?). If published, this will include your full peer review and any attached files.

**Do you want your identity to be public for this peer review?** For information about this choice, including consent withdrawal, please see our Privacy Policy.

Reviewer #1: **Yes: **Margaret Walton-Roberts

Reviewer #2: No

---

## [Decision Letter · Decision Letter 1]

14 Dec 2022

Scope of health worker migration governance and its impact on emigration intentions among skilled health workers in Nigeria

PGPH-D-22-00874R1

Dear Dr Yakubu,

We are pleased to inform you that your manuscript 'Scope of health worker migration governance and its impact on emigration intentions among skilled health workers in Nigeria' has been provisionally accepted for publication in PLOS Global Public Health.

Best regards,

Julia Robinson

Executive Editor

Reviewer Comments (if any, and for reference):

Reviewer's Responses to Questions

**Comments to the Author**

1. If the authors have adequately addressed your comments raised in a previous round of review and you feel that this manuscript is now acceptable for publication, you may indicate that here to bypass the “Comments to the Author” section, enter your conflict of interest statement in the “Confidential to Editor” section, and submit your "Accept" recommendation.

Reviewer #1: All comments have been addressed

Reviewer #2: (No Response)

2. Does this manuscript meet PLOS Global Public Health’s publication criteria? Is the manuscript technically sound, and do the data support the conclusions? The manuscript must describe methodologically and ethically rigorous research with conclusions that are appropriately drawn based on the data presented.

Reviewer #1: Yes

Reviewer #2: No

3. Has the statistical analysis been performed appropriately and rigorously?

Reviewer #1: I don't know

Reviewer #2: I don't know

4. Have the authors made all data underlying the findings in their manuscript fully available (please refer to the Data Availability Statement at the start of the manuscript PDF file)?

Reviewer #1: Yes

Reviewer #2: Yes

5. Is the manuscript presented in an intelligible fashion and written in standard English?

Reviewer #1: Yes

Reviewer #2: Yes

6. Review Comments to the Author

Reviewer #1: I am happy with the edits made, they have addressed my comments from the first round.

Reviewer #2: (No Response)

7. PLOS authors have the option to publish the peer review history of their article (what does this mean?). If published, this will include your full peer review and any attached files.

**Do you want your identity to be public for this peer review?** For information about this choice, including consent withdrawal, please see our Privacy Policy.

Reviewer #1: No

Reviewer #2: No
